# Microstructural Analysis of Fractured Orthopedic Implants

**DOI:** 10.3390/ma14092209

**Published:** 2021-04-25

**Authors:** Mateusz Kopec, Adam Brodecki, Grzegorz Szczęsny, Zbigniew L. Kowalewski

**Affiliations:** 1Institute of Fundamental Technological Research, Polish Academy of Sciences, Pawińskiego 5B, 02106 Warsaw, Poland; abrodec@ippt.pan.pl (A.B.); zkowalew@ippt.pan.pl (Z.L.K.); 2Department of Mechanical Engineering, Imperial College London, London SW7 2AZ, UK; 3Department of Orthopaedic Surgery and Traumatology, Medical University, 4 Lindleya Str, 02005 Warsaw, Poland; grzegorz.szczesny@wum.edu.pl

**Keywords:** medical fixation devices, orthopedic prostheses and implants, titanium, titanium alloy, stainless steel, microscopic fracture analysis

## Abstract

In this paper, fracture behavior of four types of implants with different geometries (pure titanium locking plate, pure titanium femoral implant, Ti-6Al-4V titanium alloy pelvic implant, X2CrNiMo18 14-3 steel femoral implant) was studied in detail. Each implant fractured in the human body. The scanning electron microscopy (SEM) was used to determine the potential cause of implants fracture. It was found that the implants fracture mainly occurred in consequence of mechanical overloads resulting from repetitive, prohibited excessive limb loads or singular, un-intendent, secondary injures. Among many possible loading types, the implants were subjected to an excessive fatigue loads with additional interactions caused by screws that were mounted in their threaded holes. The results of this work enable to conclude that the design of orthopedic implants is not fully sufficient to transduce mechanical loads acting over them due to an increasing weight of treated patients and much higher their physical activity.

## 1. Introduction

Stainless steels and titanium alloys are widely used as implant materials in orthopedic surgery due to their good biocompatibility, corrosion resistance and durability [1,2,3]. These materials are suitable for the production of biocomponents used in medicine due to their relatively high biotolerance [4], which means that they are not toxic and do not induce acute nor chronic immune reactions in organs and surrounding viable tissues [5]. One of the most relevant in vitro biocompatibility tests verifies capabilities of the living cells to multiply and migrate on the surface of implant material [6,7,8,9,10].

Austenitic chromium-nickel steel doped with molybdenum (X2CrNiMo18 14-3, also known as 316L) belongs to materials most extensively used in medicine. Nowadays, it is used to manufacture surgical instruments, surgical and cardiologic implants and joint prostheses [11,12]. Cr-Ni-Mo steels are relatively cheap and are characterized by relatively low resistance to crevice corrosion. The corrosion rate strongly depends on the properties of its passive layer, formed by thin oxide film (Cr_2_O_3_) containing small amounts of Co and Mo oxides. Their mechanical properties could be tailored during cold working [11,13]. Chromium-based steels have been widely used for the production of biocomponents [14,15,16,17,18,19].

Titanium and its alloys are well-tolerated in-between viable tissues. They are increasingly used in orthopedics as a material for production of trauma implants, including plates, screws, intramedullary nails, external fixators and joint prostheses. They are also used to manufacture surgical instruments [4]. Commercially pure titanium (Cp Ti) is believed to be among the most biocompatible metals due to its capability to form chemically stable and inert oxide layer [20]. The most important factors characterizing titanium and its alloys include low level of electric conductivity, high corrosion resistance, thermodynamic stability at physiological pH values, low tendency to form ions soluble in water and the isoelectric point of the oxide at the pH between 5 and 6. Thus, the surface protected by oxidized layer is at the physiological pH only slightly negatively charged. Titanium’s dielectric constant is comparable with that of water, which makes the Coulomb interactions of charged particles similar to that in the water [20]. Titanium alloy implants, although expensive, overpass other alloys by several advantages. They are lighter and do not corrode even in aggressive, biological environment for many years, for example, when are placed in regenerated bone with biomaterials [21]. Moreover, its physical properties, especially stiffness characterized by the Young’s modulus, resembles that of the skeletal tissue much more than steel and CoCrMo alloy. It is also not dielectric and, hence, does not warms up in electromagnetic fields enabling magnetic resonance imaging. Additionally, titanium alloys could bring to a diminished rate of bacterial colonization of prostheses, which could significantly improve the success and survival of implant-prosthetic rehabilitations on immunocompromised patients and further avoid facial perimandibular abscesses [22].

It should be mention however, that the main limitation of implant production with titanium is the difficulty in processing and making complex geometrical shapes. Additive manufacturing techniques allow to overcome such limitations. These specific techniques enable to produce functional, complex parts created directly by selectively melting layers of powder. Implants obtained from advanced additive methods, such as Direct Laser Forming and Selective Laser Melting, allow material to retain its strength properties [23,24,25,26] while adjust the shape and size individually adapted to the patient. An outstanding potentials of Additive Manufacturing processes allow to produce sophisticated, complex, 3D products dedicated for biomedical applications [27,28].

Despite the undeniable advantages of medical implants, their fractures remain a problem. Yu et al. [29] presented a retrospective clinical study performed on thirteen implants removed due to their fractures from twelve patients. The main facts observed before fracture included: screw loosening in five, marginal bone loss in five and the presence of peri-implant osteolysis in five cases. Similar retrospective studies were presented by Yi et al. [30], in which the fracture of external and internal type of implants to suggest directions for successful implant treatment was discussed. Lee et al. [31] performed a retrospective analysis of 19,006 internal stabilizations performed in 5124 patients. Stoichkov et al. [32] analyzed possible causative factors for implant fracture. In the described-above analyses, it was concluded that overload itself is a causative factor that may be responsible for implant fracture. Oh et al. [33] showed that material defects, occlusal overloads, prosthetic design and nonpassive prosthesis fit have been identified as causative factors for implant fractures. Gealh et al. [34] and Sanivarapu et al. [35] divided the potential causes of implant fracture into three categories: (1) defects in the design of the material, (2) nonpassive fit of the prosthetic structure and (3) biomechanical or physiologic overload. It should be mentioned, however, that these reported papers and factors are only some of the works reported on implant fracture [36,37,38,39,40,41]. The main factors considered as potential causes of fracture were assigned to the patient’s condition, geometry of the implant and its mechanical loads. Nevertheless, the microstructural mechanisms of fracture occurrence have not been investigated in detail as yet. Thus, we performed a microstructural analysis of the fractured trauma implants to evaluate the mechanism of the fracture leading to the implant’s damage, as well as the microstructural analysis of the alloy used for implant’s manufacturing and its composition and impurities.

## 2. Materials and Methods

Four different, exemplary implants were chosen out from 18 parts collected during last two years anteceding coronavirus pandemics to validate the observations independently from the type of an implant, its producer and an alloy itself. The study is focused on implants dedicated for the stabilization of long bone fractures, such as those that could be used temporarily, and possibly removed, when the fracture is healed; implants intended for bones subjected to mechanical loads in highest degree and implants representing materials the most often used for orthopedic purposes: steel, pure titanium and its alloy (Ti-6Al-4V).

Thus, four different parts were investigated:Pure titanium, angularly stabile, clavicular, locking plate (5.0 ChLP; ChM, Juchnowiec Kościelny, Poland; Figure 1a);Pure titanium, angularly stabile, femoral, condylar plate (7.0 ChLP, ChM, Juchnowiec Kościelny, Poland; Figure 1b);Ti-6Al-4V intramedullary nail (Triple Proximal Femoral Nail, Medgal, Poland; Księżyno, Figure 1c);X2CrNiMo18-14-3 steel femoral plate (4.5 mm VA LCP^®^Condylar Plate, DePuy Synthes, Warsaw, IN, USA; Figure 1d).

All implants served for the stabilization of bone fractures and underwent breakage during the treatment. After removal, the microstructural analysis was performed using Jeol J0L6360 LA scanning electron microscope (Jeol, Tokyo, Japan) with energy-dispersive spectroscopy (EDS) attachment (Oxford Instruments, Oxford, UK). Samples were obtained from the implant within the fracture zone. The fracture area was analyzed on both sides of the broken implant.

## 3. Results and Discussion

### 3.1. Pure Titanium Plates

The chemical composition of a titanium locking plate is presented in Table 1. The implant’s fracture zone was subjected to observations using a scanning electron microscope (Figure 2b). Figure 2a presents a general view of the fracture of exploited titanium implant with marked fracture zones characteristic for fatigue damage: (1; Figure 2a) the focus of the crack initiated around the inclusions with characteristically smoothed surface and the (2; Figure 2a) zone of fatigue breakthrough characterized by a coarse-grained structure (3; Figure 2a) residual area.

Nucleation of cracks was initiated on the side part of the implant and was followed by a progressive crack development (Figure 2b) in consequence of cyclic loading. A further microscopic analysis performed on the fracture zone showed analogical fracture characteristics throughout the sample surface, as well as the same nucleation points and fracture paths. Nucleation of the crack (4; Figure 2c), as well as their development (5; Figure 2c), can be observed. Crack development led to the material fracture due to the friction between surfaces of the crack during crack propagation. Typical fatigue behavior of the material (Figure 3b) [42] with the direction of crack propagation (6; Figure 2d), as well as visible traces of inclusions (7; Figure 2d), were observed. The mechanism of the implant’s fracture points to its mixed nature due to the presence of both plastic and brittle zones. On the fracture surface, cavities and extrusions (Figure 3a), characteristic for plastic deformation and cracks at the grain boundaries that are typical for brittle fracture, were observed (Figure 3b).

Analysis of the chemical composition of the implant material (Figure 3b, point 1) and inclusions in the implant’s fracture areas (marked with arrows, Figure 3a,b, point 2) using the EDS technique showed characteristic peaks of oxygen and aluminum, clearly indicating that the inclusions consisted of aluminum oxide Al_2_O_3_ (Figure 3 and Figure 4). These precipitates, which are characterized by extremely high hardness, serve as stress concentrators during loading. Stress components allocated around them make the crack initiation much easier in those areas rather than in homogeneous material [43]. The content of other elements in the inclusions may indicate the formation of precipitates as a result of reaction with the human body.

### 3.2. Pure Titanium Femoral Implant

The chemical composition of the titanium alloy locking plate is presented in Table 2. Visual inspection performed on the femoral implant surface (Figure 5) allows to observe numerous local plastic deformation areas of the material around the threaded holes (Figure 5a,b, areas marked with arrows). Although no material losses were found on the screws, it was assumed that they led to permanent deformation of the implant. Figure 5d presents a macroscopic view of the fracture surfaces found near the threaded hole. These fracture surfaces were characterized by high smoothness, which may indicate that long-term friction occurred between them. As the fracture of the implant was found in this particular area, scanning election microscopy was used to investigate the potential cause of failure. The SEM observations were performed on the inner part of the thread.

Figure 6 presents a general view of the crack in the inner thread of a titanium implant. The fracture was initiated in the center part of the thread (Figure 6a) and propagated deeply into the material as shown with the arrow (Figure 6b), leading to complete fracture of the implant, as shown in Figure 7. Furthermore, in the area of the dominant crack (Figure 6a, marked with the arrow), a 300-µm microcrack was observed (Figure 6c,d, marked with the arrow). Based on the observations performed on the fracture surfaces (Figure 8), it was found that, after the implant fracture, the two separated parts interacted with each other. The observed smooth surfaces (Figure 8a,b) are characteristic for friction processes, which result in surface wear of the material. The analysis of the fracture surface and the nature of the fracture allow us to state that the material was damaged due to the interaction of the implant and the screw that fastened the implant and the femur. Pieces of bone tissue were observed in the structure of the fracture, which was confirmed by the analysis of the chemical composition presented in Table 3.

### 3.3. Ti-6Al-4V Titanium Alloy

Ti-6Al-4V titanium alloy femoral gamma nail with a full set of its components (Figure 9) was subjected to visual inspection at first and, then, to microscopic observations. The chemical composition of the Ti-6Al-4V titanium alloy implant is presented in Table 4. Macroscopic examination of the implant surface allowed us to observe numerous local plastic deformations of the threads (Figure 10, marked with arrows) and the areas around the threaded holes (Figure 11, marked with arrows). These deformations caused the permanent deformation of the telescopic lag screw (part 7 in Figure 9) that was attached to the nail passing through the hole in the nail’s shaft (Figure 10a). Additionally, propagation of the longitudinal crack from the shaft towards the thread was observed on the threaded surface (Figure 10b, marked with arrow). Simultaneously, the threaded part itself was plastically deformed (Figure 10c, marked with arrow), prohibiting damage and disabling its further usage. Permanent deformation of the inner areas of the threaded holes and their locking screws are presented and marked with arrows in Figure 11a–d. A significant abrasive wear of the thread was observed, probably due to the interaction of the screw and thread itself. On the other hand, traces of material wear in the unthreaded hole were observed as a result of long-term friction (Figure 11a,b, marked with arrows) of the telescopic screw of the nail’s shaft.

Figure 12 presents an overview of the fracture area observed in the outer part of the titanium implant hole. The areas numbered 1–4 in Figure 12a during visual inspection were subjected to microscopic observations and presented in Figure 12 b–e according to numbers given. The crack was initiated at the outer part of the hole (on its edge–Figure 12b,d) and propagated deeply into the material, leading to its complete fracture. Additionally, the 1.25-mm and 0.5-mm-long cracks were observed below the main crack (Figure 12c), as well as micropores found on the edge of fractured implant (Figure 12e). Based on microscopic observations, it could be concluded that, after the implant’s break, both its separated parts rubbed on one another, producing characteristic smooth surfaces of the parts succumbing to friction. Inspection of the components of the implant and surfaces of its fracture allowed us to conclude that the implants broke in consequence of the overloads that transmitted excessive mechanical stresses from the bone to the implant’s shaft by attaching the screws.

### 3.4. X2CrNiMo18-14-3 Steel Femoral Implant

The steel femoral condylar plate (Table 5) was made of X2CrNiMo18-14-3 (D) steel (316L, PN-ISO 5832-1 Standard [44]). A macroscopic inspection (Figure 13) allowed us to observe the foci of the fracture in the area of the threaded hole (Figure 14). Smooth fracture surfaces are characteristic for the friction processes that indicated that two separated parts of the implant interacted between themselves after its break. Additionally, visible traces of wear were observed on one of the implant’s mounting screws with damage of the 35-mm-long segment of its thread.

A visual analysis (Figure 15a) revealed a characteristic crater formed around the threaded hole that occurred due to permanent deformation of the material caused by the transmission of excessive forces by the screw mounted in it. Multiple cracks propagating from the crater along the edge of the hole could also be observed (Figure 15b,c). Moreover, numerous scratches inside the nonthreaded part of the hole were observed, most likely caused by screwing the bolt into the hole or the impact of the bolt head on the side surface of the hole (Figure 15d). The edges of the second part of the fractured implant were also characterized by numerous cracks (Figure 15e–g). Analogically, smooth surfaces of the fracture edge caused by friction of one part of the broken implant over another were observed (Figure 15h).

Microscopic observations of the inner area of the fracture surface revealed dominant cracks in the inner aspect of the thread (Figure 16, marked with arrow). The crack, which initiated at the upper part of the thread, propagating deeply into the structure of the implant up to its outer surface, leading to its fracture (Figure 16a, marked with arrow). In addition, 100–300-µm microcracks could be observed on the threads and around the unthreaded parts of the hole as marked with arrows in Figure 16c,d. Bolts were subjected to an additional analysis (Figure 17), showing a significant degree of wear in one of them. The entire thread, about 30 mm long, was significantly damaged. Inspection of the fracture surface and the nature of the fracture of the implant allowed us to state that the material was damaged due to screwing it with excessive force to the plate.

A microstructural analysis of the fractured implants was performed in this paper. It should be mentioned, however, that it is hard to compare the results obtained in this work with the literature, since the knowledge on the implant fracture and its potential causes has not been reported in detail yet. More importantly, the complex loading states that the implants were subjected to were different for individual patients.

## 4. Summary and Concluding Remarks

The paper focused on the analysis of orthopedic trauma implant failure, which led to bone fracture destabilization. Implant fractures complicate orthopedic procedures, further disabling effective interventions. Clinical practice shows that they occur in a limited number of patients that usually present a fracture healing process or those who, due to unintended secondary injuries or in consequence of the transgression of postoperative recommendations, succumb to secondary limb traumatization. In those cases, implant fractures and bone break destabilization point to the causative impact of accidental or cumulative mechanical overloads of an implant.

Based on our investigations, it could be presumed that implant fractures are caused by: material impurities, where super-hard aluminum oxide acts as a mechanical strain concentrator, serving as the center of cracks propagating between the structure of an implant,the deficient adjustment of implant designs to the stresses acting on it,improper technology of implant production, especially in the case of drilling and threading of the holes serving as mounting areas of screws attaching it to the bone,inappropriate surgical technique during implantation,its mechanical overloads caused by excessive forces acting on limbs treated with this particular implant.

Nica et al. [45] analyzed the structure and remnants of implants removed due to their breaks and further reported that an inappropriate surgical technique and structural material flaws are responsible for most of the incidences of implant failures. It was concluded that the improvement of material quality and implementation of higher-quality standards would improve treatment results. The only explanation of the discrepancy of their observation with ours comes from the fact that they analyzed implants produced outside Europe.

In the materials investigated, implant fractures most probably occurred in consequence to mechanical overloads resulting from repetitive, prohibitively excessive limb loads or singular, unintended secondary injures. The first mechanism subjected the implant to relatively mild but repetitive loads, which, when summarized, damaged it. The second one fractured due to forces acting with an amplitude enormously exceeding those that the implant could withstand. Based on a finite element analysis of the broken, steel condylar plate, Gervais et al. [46] concluded that regular walks alone create mechanical forces sufficient to break the implant. The crack, initiated at the area of stress concentration due to repeated bending loads, propagated to deeper parts, resulting in the implant’s fracture. An analogous observation was reported by Hou et al. [47], where the failure mechanisms of screws manufactured by reputable producers were analyzed. The results clearly indicated that overload plays a much more important role in implant fractures than material flaws themselves.

For the observations given by our predecessors, we can add an additional one—namely, that the current design of orthopedic implants is not fully suitable to transduce mechanical loads acting on them due to the increasing weight of treated patients and, much more important, their physical activity.

Moreover, the design of implants leads to the generation of stress concentrators that serve as initiators of cracks. An analysis of the surfaces of the fractured implants revealed that, regardless of the initial material state and geometry, the fracture was caused by the concentration of the stress forces in its holes, including the threaded parts, where the crack initiated and propagated within the material. Most probably, the implants were subjected to an excessive fatigue load with additional effects caused by the interaction between the screws and threaded holes. The very tight connection between the screws and threated holes of the implant initiated cracks that led to significant wear between the working surfaces. The wear of threads of screws and plates might reduce the rigidity of the connection between the bone and the implant, thus enhancing the temporary loads between them that promoted the propagation of the implant’s cracks.

Taking into account the above observations, in order to reduce the risk of implant fracture, one can require to increase the thickness of their dimensions, especially in the areas with holes.

## Figures and Tables

**Figure 1 materials-14-02209-f001:**
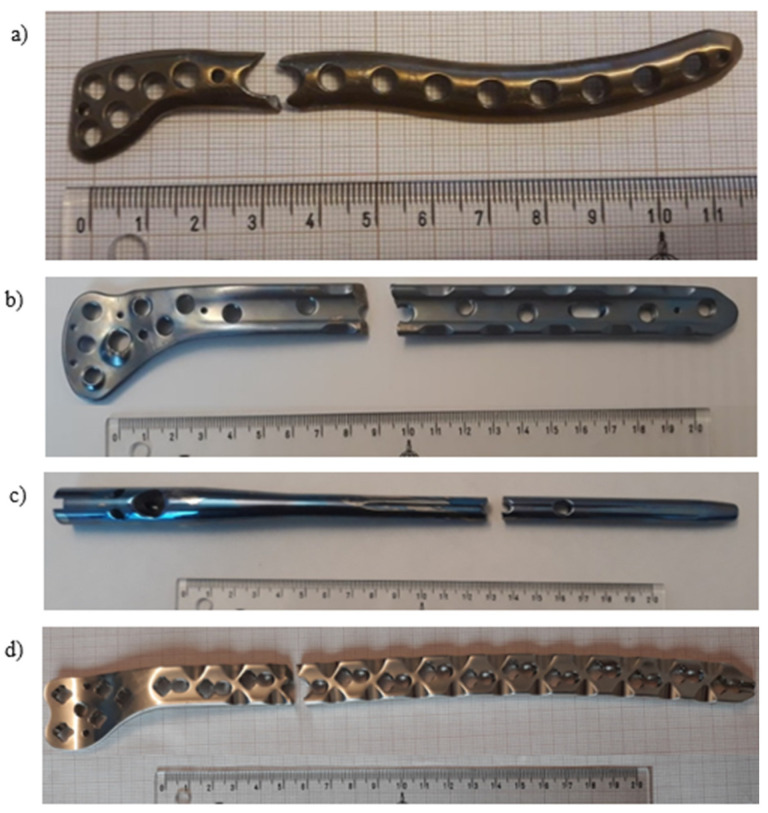
General view of the exploited implants with different geometries made of pure titanium clavicular (**a**) and distal femoral (**b**) plates, Ti-6Al-4V femoral gamma nail (**c**) and a X2CrNiMo18-14-3 femoral condylar plate (**d**).

**Figure 2 materials-14-02209-f002:**
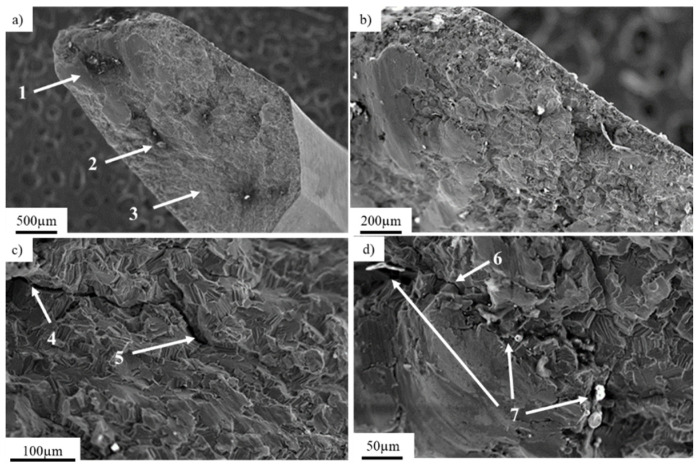
Fracture area of the titanium implant with marked fracture zones (**a**; 1–3) and its magnification showing the fatigue crack propagation zone (**b**); the crack development observed on titanium explant (**c**; 4–5) and its magnification showing the fatigue crack propagation area (**d**; 6) with local inclusions (**d**; 7).

**Figure 3 materials-14-02209-f003:**
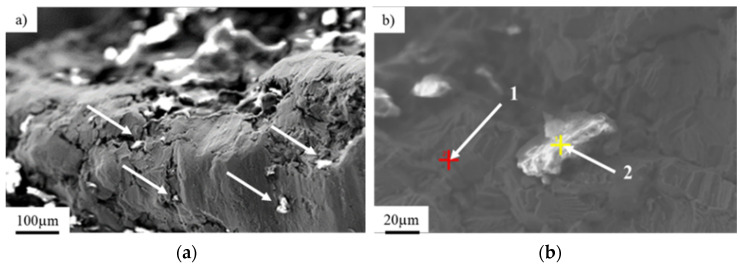
View of inclusions and precipitates on the implant surface and near the fracture zones (**a**); enlarged view of the inclusions on the implant surface (**b**).

**Figure 4 materials-14-02209-f004:**
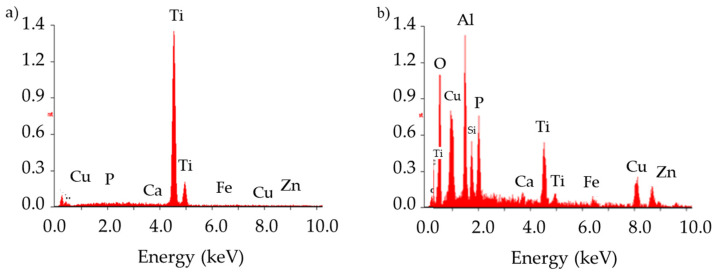
Analysis of the chemical composition of the implant (**a**) and the inclusion (**b**).

**Figure 5 materials-14-02209-f005:**
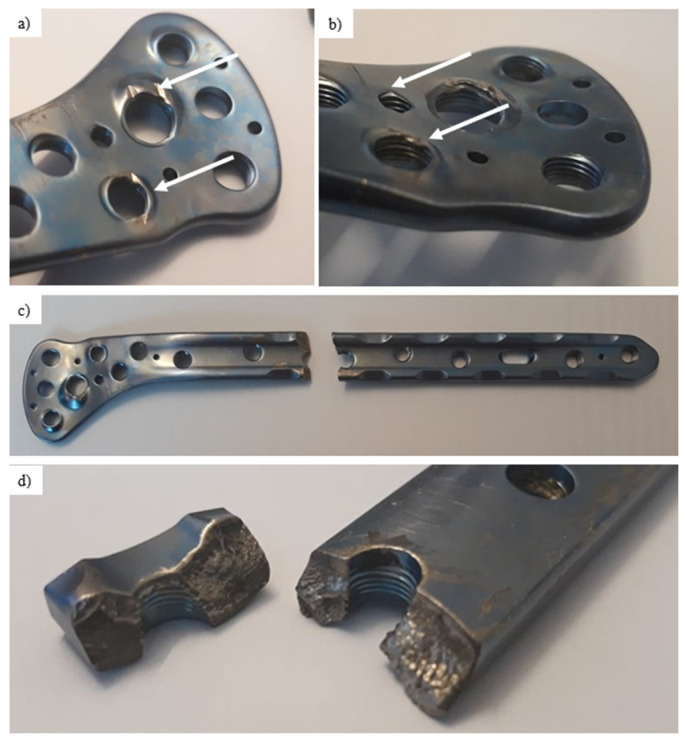
General view of the exploited titanium implant: plastic deformations found around the threated holes (**a**,**b**), fractured implant (**c**) and fracture surfaces (**d**).

**Figure 6 materials-14-02209-f006:**
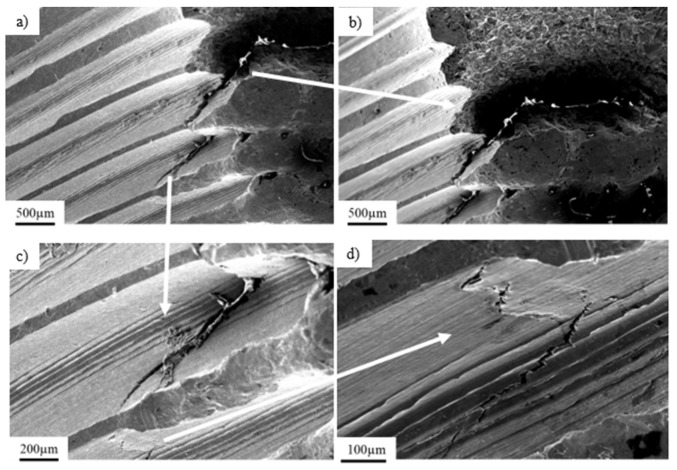
View of the dominant crack observed inside the thread (**a**), crack propagation from the thread surface (**b**), enlarged area of the crack initiation (**c**) and surface cracks observed near the dominant crack (**d**).

**Figure 7 materials-14-02209-f007:**
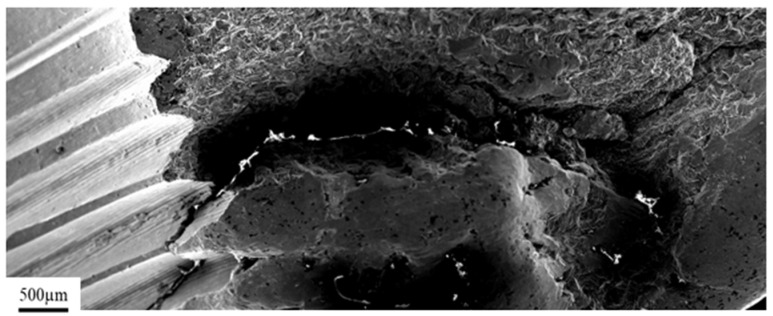
Crack propagation initiated in the center part of the thread.

**Figure 8 materials-14-02209-f008:**
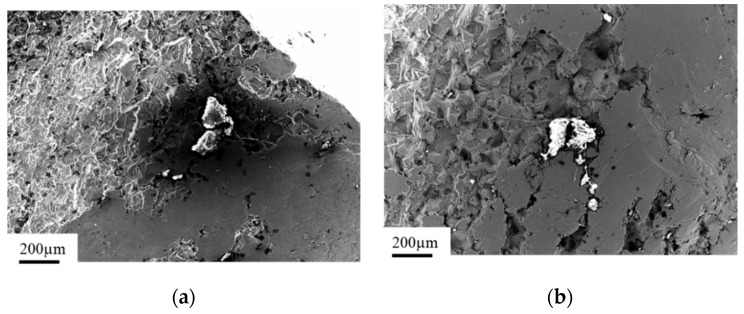
Fracture surfaces observed on both parts of the fractured implant: left part (**a**); right part (**b**).

**Figure 9 materials-14-02209-f009:**
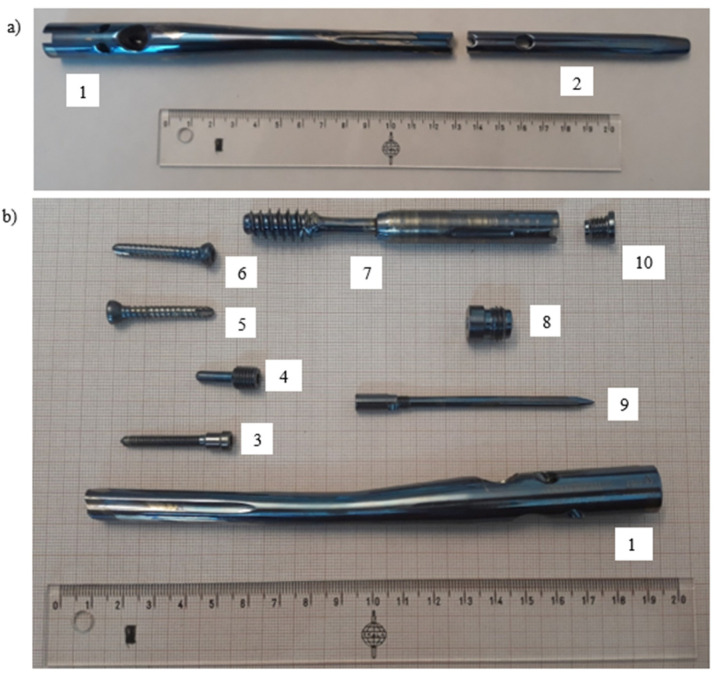
General view of the broken femoral nail’s shaft; its proximal (1) and distal (2) parts (**a**) with a set of assembling parts: proximal part of the broken nail’s shaft (1); blocking bolt (3); nail’s screw (4); two self-tapping locking screws (5 and 6); telescopic lag screw (7); end cap (8); anti-rotation pin (9); cap (10) (**b**).

**Figure 10 materials-14-02209-f010:**
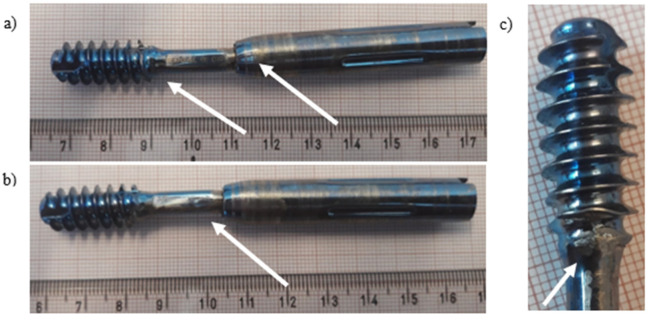
Visual inspection of the exploited telescopic lag screw (number 7 in Figure 9b) of the titanium implant: multiple cracks found on both sides of the telescopic lag screw (**a**,**b**); broken thread (**c**).

**Figure 11 materials-14-02209-f011:**
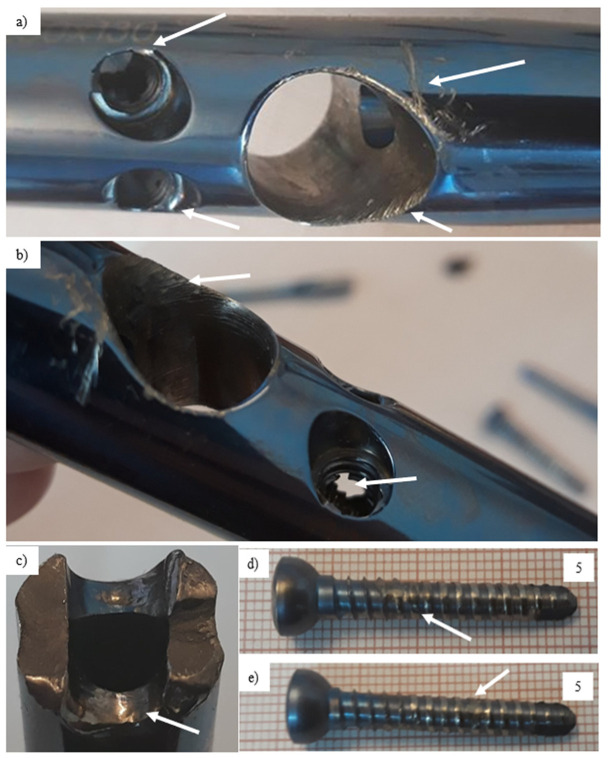
Visual inspection of the exploited part (number 1 in Figure 9a) of the titanium implant: worn marks (**a**); broken hole thread (**b**); fracture surface with area of wear (**c**); broken threads of screws (**d**,**e**) (number 5 in Figure 9b).

**Figure 12 materials-14-02209-f012:**
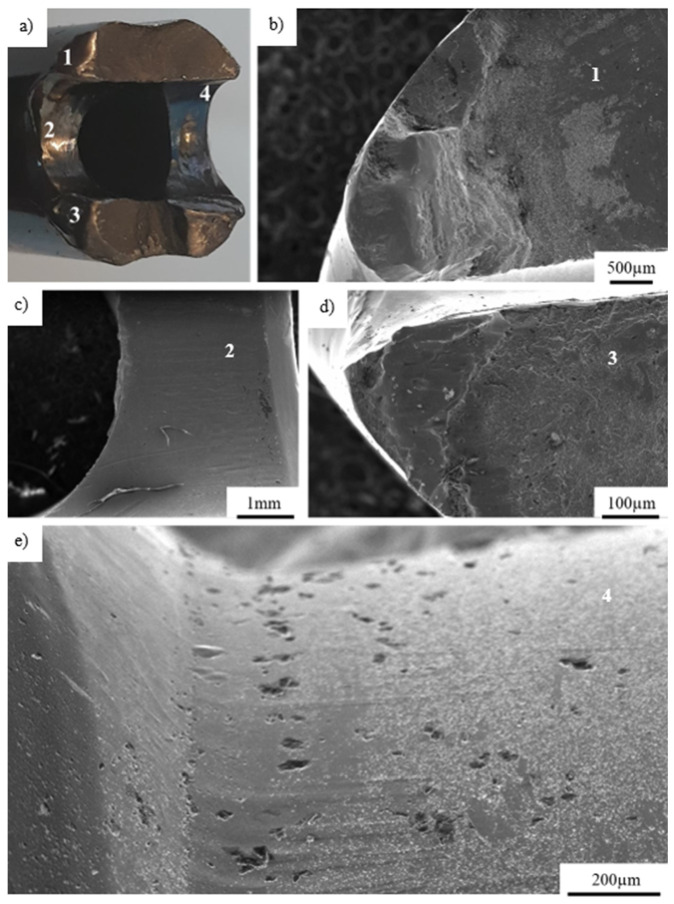
Macroscopic view of the fracture surface with marked areas of SEM inspection (**a**); the edge of fractured implant (**b**); cracks observed in the central part of hole (**c**); the second edge of fractured implant (**d**); micropores found on the edge of the fractured implant (**e**).

**Figure 13 materials-14-02209-f013:**
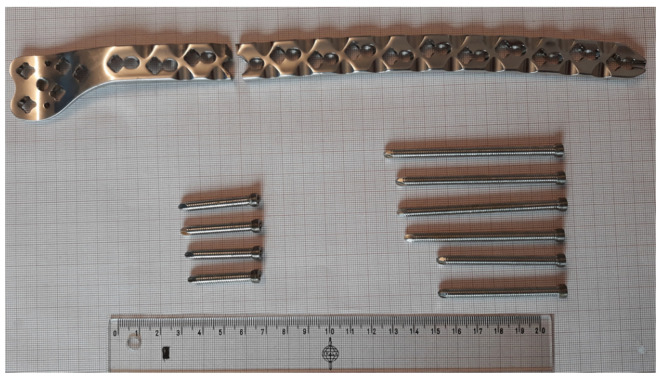
General view of the explanted steel femoral condylar plate.

**Figure 14 materials-14-02209-f014:**
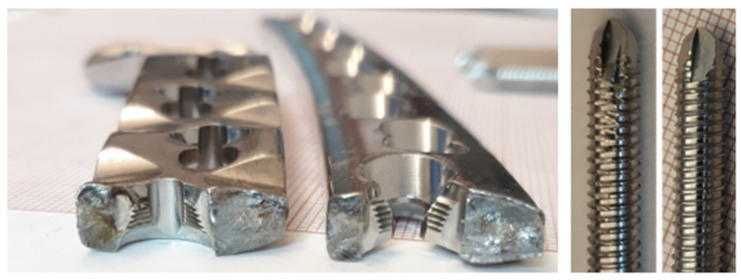
Visual inspection of the fracture surface of the steel implant compared to the as-received and exploited screw.

**Figure 15 materials-14-02209-f015:**
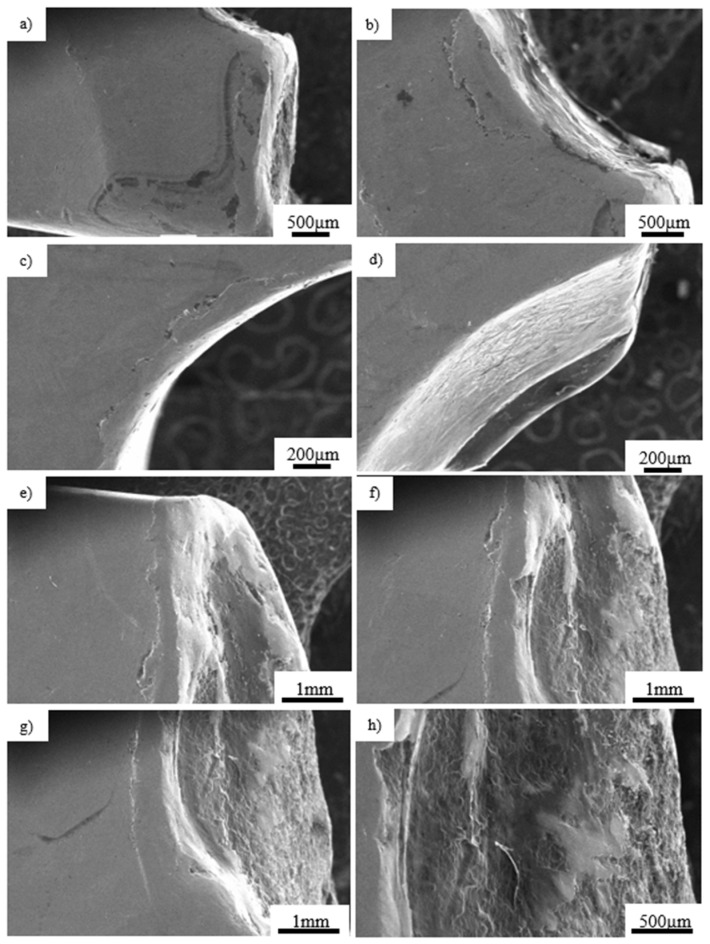
Microscopic observations of the fracture surface of the steel implant: crater formed around the threaded hole (**a**); cracks propagating from the crater along the edge of the hole (**b**,**c**); scratches inside the nonthreaded part of the hole (**d**); numerous cracks on edges of the second part of the fractured implant (**e**–**g**); smooth surfaces of the fracture edge caused by friction of one part of the broken implant over another (**h**).

**Figure 16 materials-14-02209-f016:**
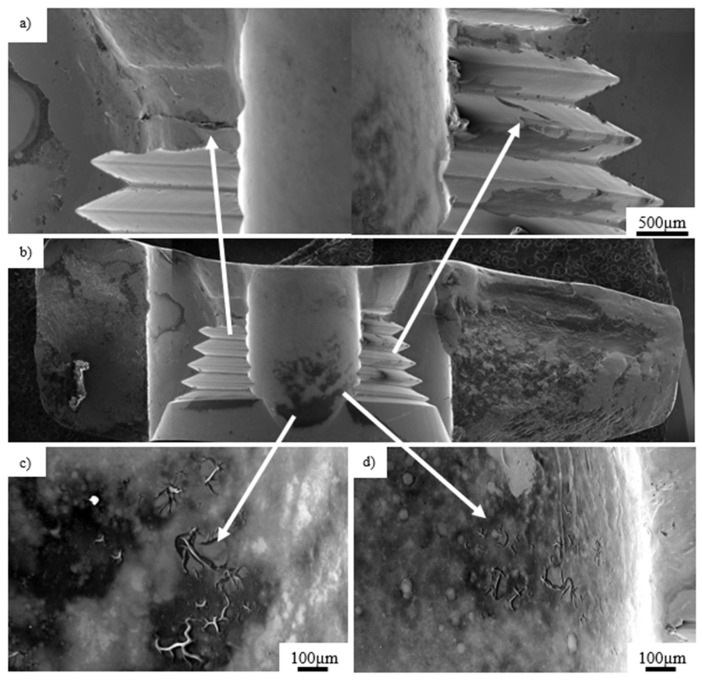
Microscopic observations of the inner area of the fracture surface.

**Figure 17 materials-14-02209-f017:**
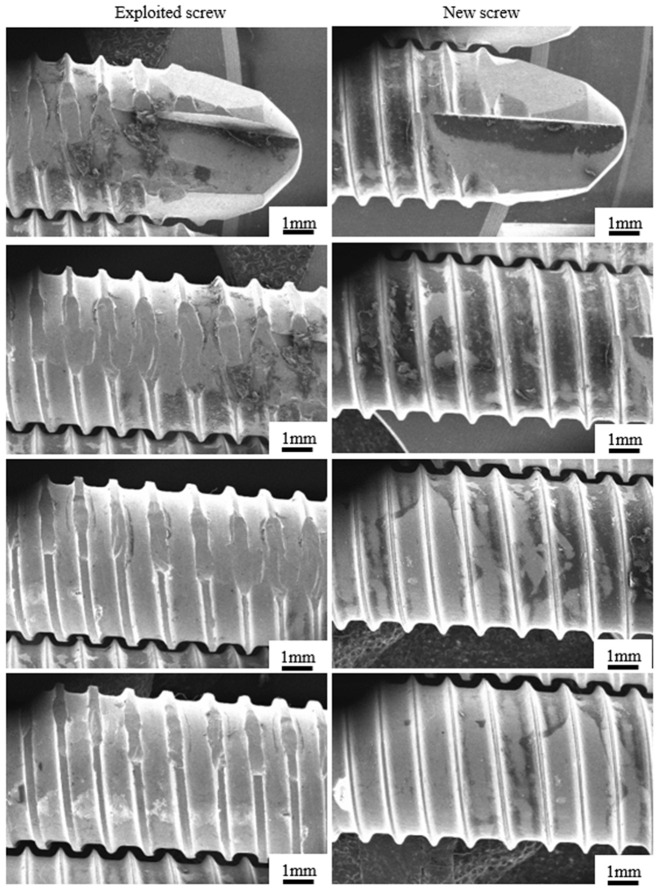
Comparison of the explanted screw (left column) and the new one (right column).

**Table 1 materials-14-02209-t001:** Chemical composition of the titanium locking plate.

Element	Ti	Impurities
Wt %	99.20	00.80

**Table 2 materials-14-02209-t002:** Chemical composition of the pure titanium femoral implant.

Element	Ti	Impurities
Wt %	99.12	00.88

**Table 3 materials-14-02209-t003:** Chemical composition of the intrusions found in the fracture surface of the titanium femoral implant.

Element	C	O	P	Mo	Cl	K	Ca	Ti
Wt %	17.34	04.77	12.65	02.27	01.40	01.56	51.95	08.04

**Table 4 materials-14-02209-t004:** Chemical composition of the titanium alloy femoral/pelvic implant.

Element	Ti	Al	V
Wt %	90.12	06.72	03.16

**Table 5 materials-14-02209-t005:** Chemical composition of the steel femoral implant.

Element	C	Si	Mo	Cr	Fe	Ni
Wt %	2.11	0.64	3.18	17.45	61.73	14.89

## Data Availability

The data are available in a publicly accessible repository.

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
