# Peer review of "Microstructural Analysis of Fractured Orthopedic Implants"

_materials, 2021, doi:10.3390/ma14092209_

Round 1
Reviewer 1 Report
The paper is well written and the topic represents an important concern about orthopedics but also titanium implants used for oral rehabilitation. Indeed, results presented by authors can lead to new applications of titanium alloy to improve mechanical behavior of in implantology. the materials and methods are adequate and rigorous, results are clearly described and fit very well with proposed objectives.
The weakness of the paper is linked to the introduction and discussion, where Authors should improve some sentences about potential link with clinical topics and potential clinical application (e.g. in dentistry). Such aspect can open presented results to a wide range of readers and also trade (technological implications in the field of dentistry. Indeed, Authors should include in their references some papers about the following crucial clinical points. The new alloy can improve mechanical resistance of implants when applied in particular clinical situations, for example when are placed in regenerated bone with biomaterials (PubMed ID: 27496576; ) The other very important clinical application is the hypothesis that particular alloys of titanium can bring to a diminished rate of bacterial colonization of prostheses, that could significantly improve the success and survival of implant-prosthetic rehabilitations on immunocompromised patients, in endodontics (PubMed ID: 18811596), avoiding facial perimandibular abscesses (PubMed ID: 19821124).
After these changes the paper can be considered for publication
Author Response
Detailed Response to Reviewer Comments
Ms. Ref. No.: materials-1176401
Title: Microstructural analysis of fractured orthopedic implants
Materials Dear Sir or Madame,
I would like to thank you very much for your letter and the reviewer’s comments on our manuscript (No.: materials-1176401). We appreciate your very valuable comments, that gave us a chance for revising the manuscript.
We have addressed all of the comments and revised the manuscript accordingly. All of the changes have been highlighted in yellow in the revised manuscript. Detailed responses to the comments are described in the “Response to Reviewers” point by point.
We now resubmit the manuscript for your further consideration for publication in your journal. We sincerely hope this revised manuscript will be finally acceptable for publication. If you have any questions about this manuscript, please do not hesitate to contact me.
Best regards
Mateusz Kopec
On behalf of all co-authors
Institute of Fundamental Technological Research
Polish Academy of Sciences
E-mail: mkopec@ippt.pan.pl
Reviewer’s Comments:
Reviewer #1:
The paper is well written and the topic represents an important concern about orthopedics but also titanium implants used for oral rehabilitation. Indeed, results presented by authors can lead to new applications of titanium alloy to improve mechanical behavior of in implantology. the materials and methods are adequate and rigorous, results are clearly described and fit very well with proposed objectives.
The weakness of the paper is linked to the introduction and discussion, where Authors should improve some sentences about potential link with clinical topics and potential clinical application (e.g. in dentistry). Such aspect can open presented results to a wide range of readers and also trade (technological implications in the field of dentistry. Indeed, Authors should include in their references some papers about the following crucial clinical points. The new alloy can improve mechanical resistance of implants when applied in particular clinical situations, for example when are placed in regenerated bone with biomaterials (PubMed ID: 27496576; ) The other very important clinical application is the hypothesis that particular alloys of titanium can bring to a diminished rate of bacterial colonization of prostheses, that could significantly improve the success and survival of implant-prosthetic rehabilitations on immunocompromised patients, in endodontics (PubMed ID: 18811596), avoiding facial perimandibular abscesses (PubMed ID: 19821124).
Response: We would like to thank the reviewer for the comment. Corrections were highlighted in the article and a detailed response is located below.
Titanium alloy implants, although expensive, overpass other alloys by several advantages. They are lighter and do not corrode even in highly aggressive, biological environment for many years, for example when are placed in regenerated bone with biomaterials [21]. Moreover, its physical properties, especially stiffness characterized by the Young’s modulus, resembles that of the skeletal tissue much more than steel and CoCrMo alloy. It is also not dielectric, and hence, does not warms up in electromagnetic fields enabling magnetic resonance imaging. Additionally, titanium alloys could bring to a diminished rate of bacterial colonization of prostheses, which could significantly improve the success and survival of implant-prosthetic rehabilitations on immunocompromised patients and further avoid facial perimandibular abscesses [22].

Reviewer 2 Report
The authors address an interesting research topic for the Materials Journal readership.
It is a well-organized and relatively well-written paper, that is why I uphold its publication in the Journal.
However, there are some deficiencies, which should be considered for the benefits of the readership, the reputation of the Journal, and the authors themselves.
First, there are some issues about the paper content. The dubious or questionable passages, which I mark in blue in the attached marked manuscript MarkedReview.materials-1176401.pdf. I will comment in these notes below.
Besides, although the English is readable and understandable, there are some issues worthy of correction or improvement. I address them directly by marks, comments and suggestions, made in red in the manuscript attached herewith.
Well, apropos the content, I have general comment about the figures. Most readers limit to look over the abstract, the figures, and the conclusions. Then, if the authors aim to achieve any impact, and for the readership benefit, it is advisable to make the figures self-sufficient, I mean, containing everything to make the things clear at a glance. In particular, the figure captions must explain the meanings of all arrows, number marks, etc.ç
The sentence in the lines 47-50 mentions, among other things, the “thermodynamic state”. I do not catch what do you mean here specifically – what state characteristics do you refer to.
Next, the passage in the lines 119-137 is confusing. In particular, the text in the lines 129-132, I guess, deciphers the meanings of numeric marks in Fig. 2, but not in Fig. 3. If I’m right, it is advisable to terminate all the things about Fig. 2 in the first paragraph of this passage, and then turn to the commenting of Fig. 3. Besides, I do not understand what do you mean as “faults” in the line 136: this is too generic word, but in the field of failure analysis other more specific terms are in common use. Finally, I do not see, what relation the ref. [40] has to Fig. 3.
Abbreviations, like EDS in the line 143, must be deciphered at the first use.
About the hardness value in the line 145, you do not report any hardness measurements. Where does this value come from? Please, justify the magnitude and provide the source of data.
Please, clarify the meaning of “the stiffness of the connection” in the line 333. In mechanics, stiffness is the ratio between the force and the displacement caused by it. But I failed to see what both these components are in the present context.
To summarise, I recommend this paper for publication after suggested improvements. Some of them are really necessary.

Author Response
Detailed Response to Reviewer Comments
Ms. Ref. No.: materials-1176401
Title: Microstructural analysis of fractured orthopedic implants
Materials
Dear Sir or Madame,
I would like to thank you very much for your letter and the reviewer’s comments on our manuscript (No.: materials-1176401). We appreciate your very valuable comments, that gave us a chance for revising the manuscript.
We have addressed all of the comments and revised the manuscript accordingly. All of the changes have been highlighted in yellow in the revised manuscript. Detailed responses to the comments are described in the “Response to Reviewers” point by point.
We now resubmit the manuscript for your further consideration for publication in your journal. We sincerely hope this revised manuscript will be finally acceptable for publication. If you have any questions about this manuscript, please do not hesitate to contact me.
Best regards
Mateusz Kopec
On behalf of all co-authors
Institute of Fundamental Technological Research
Polish Academy of Sciences
E-mail: mkopec@ippt.pan.pl
Reviewer’s Comments:
Reviewer #2:
The authors address an interesting research topic for the Materials Journal readership.
It is a well-organized and relatively well-written paper, that is why I uphold its publication in the Journal.
However, there are some deficiencies, which should be considered for the benefits of the readership, the reputation of the Journal, and the authors themselves.
First, there are some issues about the paper content. The dubious or questionable passages, which I mark in blue in the attached marked manuscript MarkedReview.materials-1176401.pdf. I will comment in these notes below.
Besides, although the English is readable and understandable, there are some issues worthy of correction or improvement. I address them directly by marks, comments and suggestions, made in red in the manuscript attached herewith.
Response: We would like to thank the reviewer for the comments. Corrections were made accordingly and highlighted in the article.
Well, apropos the content, I have general comment about the figures. Most readers limit to look over the abstract, the figures, and the conclusions. Then, if the authors aim to achieve any impact, and for the readership benefit, it is advisable to make the figures self-sufficient, I mean, containing everything to make the things clear at a glance. In particular, the figure captions must explain the meanings of all arrows, number marks, etc.ç
Response: We would like to thank the reviewer for the comments. Corrections were made and additional information were add. The captions were prepared in accordance to the arrows or numbers on figures.
The sentence in the lines 47-50 mentions, among other things, the “thermodynamic state”. I do not catch what do you mean here specifically – what state characteristics do you refer to.
Response: We would like to thank the reviewer for the comment. The authors meant “thermodynamic stability at physiological pH values”. Word state was misused.
Next, the passage in the lines 119-137 is confusing. In particular, the text in the lines 129-132, I guess, deciphers the meanings of numeric marks in Fig. 2, but not in Fig. 3. If I’m right, it is advisable to terminate all the things about Fig. 2 in the first paragraph of this passage, and then turn to the commenting of Fig. 3. Besides, I do not understand what do you mean as “faults” in the line 136: this is too generic word, but in the field of failure analysis other more specific terms are in common use. Finally, I do not see, what relation the ref. [40] has to Fig. 3.
Response: We would like to thank the reviewer for the comment. Corrections were highlighted in the article. Detailed response is located below.
On the fracture surface, cavities and extrusions (Figure 3a), characteristic for plastic de-formation and cracks at the grain boundaries that are typical for brittle fracture, were observed (Figure 3b).
Abbreviations, like EDS in the line 143, must be deciphered at the first use.
Response: We would like to thank the reviewer for the comment. Information about EDS was added to materials and method section.
About the hardness value in the line 145, you do not report any hardness measurements. Where does this value come from? Please, justify the magnitude and provide the source of data.
Response: We would like to thank the reviewer for the comment. This value is not relevant for the paper thus was delated.
Please, clarify the meaning of “the stiffness of the connection” in the line 333. In mechanics, stiffness is the ratio between the force and the displacement caused by it. But I failed to see what both these components are in the present context.
Response: We would like to thank the reviewer for the comment. Authors meant rigidity of the connection. Word stiffness was misused.
To summarise, I recommend this paper for publication after suggested improvements. Some of them are really necessary.
Response: Thank you very much for your valuable comments.
